# Developing an Analogue Residual Limb for Comparative DVC Analysis of Transtibial Prosthetic Socket Designs

**DOI:** 10.3390/ma13183955

**Published:** 2020-09-07

**Authors:** Kathryn Rankin, Joshua Steer, Joshua Paton, Mark Mavrogordato, Alexander Marter, Peter Worsley, Martin Browne, Alexander Dickinson

**Affiliations:** 1Bioengineering Science Research Group, School of Engineering, University of Southampton, Southampton SO17 1BJ, UK; k.rankin@soton.ac.uk (K.R.); joshua.steer@soton.ac.uk (J.S.); jtap1u17@southamptonalumni.ac.uk (J.P.); a.d.marter@soton.ac.uk (A.M.); doctor@soton.ac.uk (M.B.); 2µ-VIS X-Ray Imaging Centre, University of Southampton, Southampton SO17 1BJ, UK; mnm100@soton.ac.uk; 3Skin Health Research Group, School of Health Sciences, University of Southampton, Southampton SO16 6YD, UK; p.r.worsley@soton.ac.uk; 4Institute for Life Sciences, University of Southampton, Southampton SO17 1BJ, UK

**Keywords:** amputation, prosthetic, digital volume correlation, DVC, micro-CT

## Abstract

Personalised prosthetic sockets are fabricated by expert clinicians in a skill- and experience-based process, with research providing tools to support evidence-based practice. We propose that digital volume correlation (DVC) may offer a deeper understanding of load transfer from prosthetic sockets into the residual limb, and tissue injury risk. This study’s aim was to develop a transtibial amputated limb analogue for volumetric strain estimation using DVC, evaluating its ability to distinguish between socket designs. A soft tissue analogue material was developed, comprising silicone elastomer and sand particles as fiducial markers for image correlation. The material was cast to form an analogue residual limb informed by an MRI scan of a person with transtibial amputation, for whom two polymer check sockets were produced by an expert prosthetist. The model was micro-CT scanned according to (i) an unloaded noise study protocol and (ii) a case study comparison between the two socket designs, loaded to represent two-legged stance. The scans were reconstructed to give 108 µm voxels. The DVC noise study indicated a 64 vx subvolume and 50% overlap, giving better than 0.32% strain sensitivity, and ~3.5 mm spatial resolution of strain. Strain fields induced by the loaded sockets indicated tensile, compressive and shear strain magnitudes in the order of 10%, with a high signal:noise ratio enabling distinction between the two socket designs. DVC may not be applicable for socket design in the clinical setting, but does offer critical 3D strain information from which existing in vitro and in silico tools can be compared and validated to support the design and manufacture of prosthetic sockets, and enhance the biomechanical understanding of the load transfer between the limb and the prosthesis.

## 1. Introduction

For the estimated 0.5% of the world’s population who need prosthetic or orthotic services [1], successful rehabilitation has a substantial influence on their quality of life. Lifetime healthcare costs are substantial for individuals with amputation, estimated to average nearly £3000/yr per patient [2]. This includes considerable expenditure on prosthetic limb components: a child losing a limb at 10 years old may require 25 prosthetic limbs in their lifetime [3]. A critical component of the prosthetic limb is the socket, which is the load-bearing interface between the prosthesis and the residual limb. The socket plays the primary role in providing a sound, functional coupling allowing transfer of dynamic loading from gait and other activities whilst distributing these loads to the skin, soft tissues and musculoskeletal system, to minimise discomfort and the risk of tissue damage.

Prosthetic sockets are personalised to accommodate different residual limb anatomy, through design and fabrication by expert prosthetists. Several well-established design theories exist, including the Total Surface Bearing (TSB) approach [4], which aims to minimise pressure gradients across the limb–socket interface, and the Patellar Tending Bearing (PTB)) approach [5,6], which preferentially loads the residual limb’s pressure tolerant zones, and unloads sensitive areas, such as the fibular head. Prosthetists often design and fit the socket by an iterative process informed by their client’s feedback, and observation of physiological responses to pressure, for example, skin blanching. Socket fit is described as a primary concern of many prosthesis users [7], but discomfort and soft tissue damage are common [8], especially during early rehabilitation. Indeed, prosthesis users need numerous visits to clinics, returning on average nine times in their first year post-amputation across a range of different amputation levels and socket types [9,10]. Whilst some iterative work is inevitable as the residual limb reduces in volume during post-surgical healing and biomechanically adapts, the clinical prosthetics community’s consensus is that objective tools are needed to enhance the evidence base for socket design, supporting the application of their experience and skill, and reducing avoidable iterations.

Researchers have presented tools to assess the mechanical conditions at the skin–socket interface using sensors [11,12] and the use of finite element (FE) analysis to predict these conditions to inform socket design was first proposed in the 1980s [13,14]. Recent work has established highly computationally efficient implementations, with a view to facilitate translation into the clinic [15]. Previously, it has only been possible to validate an FE model’s surface stress predictions [16], but it is recognised that soft tissue injury is a product of tissue shear strains in the superficial and in deeper tissues, particularly around bony prominences [17]. These tissue shear strains may not correspond to surface pressure points or gradients, thus requiring in vivo or in silico estimation of tissue deformations. In the latter case, there is a need to provide confidence in finite element (FE) models’ volume strain predictions, which could be addressed through digital volume correlation (DVC).

DVC involves estimating a displacement field between two volumetric imaging datasets, often reconstructed from micro-focus computed tomography (µCT), by correlating unique pattern features in subvolumes in the voxel intensity array [18]. In contrast to surface-based digital image correlation, which often employs an applied pattern of speckles, DVC typically uses features in a material’s structure such as the reinforcement phase of composites [19] or the trabecular rod, plate and pore structures of trabecular bone [20,21], as well as microstructural features in cortical bone [22,23,24].

Soft tissue strain fields have proven harder to characterise using these techniques, owing to a lack of highly-contrasting pattern in µCT imaging [25]. Researchers have tracked mitral valve deformations in µCT by applying glass, sand, sea-shell and clay fiducial markers to the tissues’ surface [26]. Other researchers have approached the problem using different imaging modalities, for enhanced contrast. The strain field in tendon fascicles has been estimated with DVC using tenocyte nuclei as the pattern, stained for increased contrast, and imaged using confocal microscopy [27]. Optical coherence tomography (OCT) may allow larger structures to be imaged than confocal microscopy, but still only within superficial tissue samples (millimetre depth penetration). This has been applied in 3D to a variety of ex vivo and phantom materials containing particles [28,29]. More recently, DVC has been applied to cadaveric intervertebral discs using native tissue features visible using ultra high-field (9.4 T) MRI [30], and using ECG-gated CT angiographic imaging for in vivo characterisation of aortic wall displacement and strain [31]. The translation of µCT imaging-based DVC is limited in the in vivo clinical setting due to radiation dose and lack of contrasting pattern for correlation. However, phase contrast µCT imaging has been proposed as a promising future direction for visualising the required texture in ex vivo soft tissues for strain field estimation [25], recently demonstrated for the cartilage–subchondral bone interface [32]. 

We propose that DVC may be used to gain a deeper understanding of the sub-surface load transfer from prosthetic sockets into the residual limb, to compare socket design parameters and support FE model validation. Therefore, this study’s aim was to develop an analogue transtibial amputated residual limb for volumetric strain estimation using DVC and assess its sensitivity to detect the strain patterns typically observed in residual limbs during prosthetic socket use.

## 2. Materials and Methods 

### 2.1. Soft Tissue Analogue Material

First an analogue material was developed, to represent the gross stiffness of residual limb soft tissues at small strains. This included a distribution of fiducial markers enabling the images to be used in DVC analysis. The target stiffness was set around 100 kPa. Transtibial residual limb soft tissues are reported to present a range of stiffness from approximately 50–150 kPa, or 50–100 kPa excluding the patella tendon [14,33,34], reaching the higher end of this range when muscles are contracted. Samples of a range of two-part silicone elastomers were prepared, including examples with deadener additive to reduce the nominal stiffness, and the addition of sand particles in varying volume fractions. The specimens were cured under −1 bar vacuum to minimise porosity introduced in mixing. The materials’ compressive modulus was characterised according to Standard ASTM D 575-91 [35]. The tests were performed on an Instron 5569 mechanical test machine (Instron Corp, Norwood, MA, USA) with a 2kN load cell, in ramped displacement-controlled compression up to 60% strain at 5%/sec, between flat plates lubricated with bicycle oil to allow transverse expansion. The compressive force was used to calculate engineering- and estimated true stress, assuming incompressibility [36], and plotted against strain calculated from the loading plate displacement, which was corrected for machine compliance. These data were used to estimate linear tangent moduli. The selected material included a silicone rubber matrix of 62.5% Platsil Gel 25 and 37.5% Platsil Gel 25 Deadener LV (Mouldlife, Bury Saint Edmunds, UK), giving a tangent modulus from 0–10% strain of 74 kPa (R^2^ = 0.974). This increased to 95 kPa (R^2^ = 0.990) when mixed with 0.1–0.5 mm size range sand (EuroSand, Ziegler Group, Weiden, Germany) at approximately 15% by volume (30% by mass).

### 2.2. Analogue Residual Limb Fabrication

MRI scans of a single, left transtibial amputated residual limb were obtained under secondary data analysis ethical approval (ERGO29927). The participant (male, 67 yrs) had a unilateral transtibial amputation 12 years earlier due to complications from diabetes mellitus. Prior to imaging, the participant provided written, informed consent (Fraunhofer IPA #2016_BLM_0009). The scans were collected in a 3 T MRI scanner (MAGNETOM Spectra, Siemens Healthcare GmBH, Erlangen, Germany at Radiologie im Zemtrum, Agusburg, Germany), with T1-weighting using a spine coil, 13 ms echo time, and 742 ms repetition time, producing 3.0 mm axial slices and 0.5 × 0.5 mm in-plane resolution (Figure 1A). The participant was scanned whilst wearing his usual elastomeric prosthetic liner. The MRI image stack was segmented (ScanIP 2017.06, Synopsys Inc., Mountain View, CA, USA) to provide representations of the liner’s external surface, and the residual bones. The reconstructed MRI was used to identify nearest size-match composite bone models (4th Generation Composite Sawbones, Femur #3403, Tibia #3401, Fibula #3427-1 and Patella #3419 with tabs removed, Sawbones AG, Malmö, Sweden), which were resected and potted in resin (Technovit 3040, Kulzer GmbH, Wehrheim, Germany). This included simplified collateral ligaments and rubber sheet, to provide nominal compliance at the articulating surfaces (Figure 1B). A nylon mould was 3D printed to the residuum’s neutral shape. The same Platsil25 and sand mixture was stirred, degassed at −1 bar for 4 min, poured into the mould (which contained the residual bones supported by a retort stand), and degassed again at −1 bar for 2.5 min before being left to cure (Figure 1C). The model was extracted from the mould and a nylon stocking was donned, with a polyethylene terephthalate (PETE) check socket manufactured for the study participant by an expert prosthetist (Figure 1D,E). The assembly was mounted in a load frame comprising steel plates, carbon fibre pillars, and a single-axis actuator (Deben UK Ltd., Bury St Edmunds, UK, Figure 1F), with a 1 kN capacity and 10 mm travel. The bone-soft tissue surface alignment was matched to the MRI scans. In the test rig, the loading axis was aligned between the cut end of the femur and the distal-most tip of the socket. The socket was loaded through a shallow cylinder, which allowed socket rotation but constrained off-axis translation of the socket tip.

### 2.3. Digital Volume Correlation Protocol

The assembly was mounted on the rotate stage within a custom 225/450 kVp Nikon/X-tek Hutch micro-focus X-ray CT scanner (Nikon Metrology, Tring, UK, at the µ-VIS X-ray Imaging Centre, University of Southampton, Figure 2). µCT scans were conducted using the 450 kVp source and Perkin Elmer XRD 1621 CN03 HS detector (PerkinElmer Optoelectronics, Wiesbaden, Germany), with a source-to-object distance of 440.9 mm and a source-to-detector distance of 800.7 mm. The polychromatic X-ray beam had a 320 kVp peak voltage, with 59.2 W power. 3142 projection images were acquired using 12 dB analogue gain and 134 ms exposures, averaging four frames/projection through 360 degrees rotation, taking 30 min in total. Four scans were conducted for an unloaded noise study, and four to compare the soft tissue analogue strain fields generated in TSB and PTB design sockets under a static axial displacement corresponding with 400 N load, representing two-legged stance (Table 1). Following preliminary tests, it was identified that a ~20 min dwell period was required between loading and scanning, for viscoelastic stress relaxation effects to subside. A pre-determined displacement was applied to give the desired load after stress relaxation.

The projection images were used to reconstruct 3D volumes of 2000 × 2000 × 2000 voxels (cubic pixels) using CTPro3D (Nikon Metrology, Tring, UK) with 108 µm voxel size. DVC was performed using combined fast-Fourier transform (FFT) and direct correlation (DC) algorithms (DaVis 8, LaVisionUK Ltd., Bicester, UK). The DVC parameters were verified by performing multiple correlation attempts between unloaded noise study image pairs with subvolume sizes from 28–128 vx, and 50% overlap. Using the verified subvolume size, DVC correlations were performed between the unloaded reference and loaded scans with each socket, in four steps with two passes, producing three directional displacements and six strain components. The results data points were filtered to remove those outside the residuum model, and within the bone structures. Spatial strain distribution histograms and percentile ranges were calculated.

## 3. Results

### 3.1. Noise Study

The sand particles and silicone gel showed high contrast in the reconstructed µCT images (Figure 3). The particles were well-distributed throughout the structure, except for some sparse regions within the knee joint space and a dense space at the distal tip, where ~5mm particle sinking before full curing caused particle collection. CT cone beam artefacts were visible at the proximal and distal ends of the scan, outside the main region of interest. Of the tested subvolume sizes, the optimal balance of strain sensitivity vs. spatial resolution of strain was obtained with 64 vx DVC subvolumes overlapping at 50% (Figure 4, Appendix A
Table A1). In the three nominally-uniform strain scan pairs, with 64 vx subvolumes all strain components had a standard deviation of strain below 0.32% (translation case), 0.13% (magnification case), and 0.02% (repeat case). This 64 vx subset with 50% overlap and 108 µm voxel size represents a spatial resolution of strain of approximately 3.5 mm.

### 3.2. Comparative Socket Design Study

Axial and shear strain fields were extracted for the stance-loaded sockets when compared to their donned and unloaded reference case, which are plotted on example sagittal and coronal sections through the main axis of the tibia (Figure 5). The strain fields were analysed quantitatively by calculating spatial strain distribution histograms (Figure 6) and percentile ranges (Appendix A
Table A2). In a region of interest (ROI) containing all soft tissue below the tibial plateau, axial strain magnitudes were typically in the range of −4 to +6%, and shear strain ranged from −10 to +9% (10th–90th) percentile ranges. There were many similarities in the sockets’ load transfer mechanisms:

Both sockets demonstrated relatively uniform tensile and shear strain fields around the tibia, and compression below the tibial plateau flare (Figure 5 ‘’tpf’’), visible on the coronal plane. Relatively low distal soft tissue loading was observed for both sockets, with median 0.17% (10th–90th percentile range −2.52–2.89) in a distal ROI below the tibial tip in the coronal plane for the TSB socket, and −0.13% (−3.92–1.72) for the PTB socket. This distal soft tissue protection was achieved through the sockets’ press-fit, which generates the observed shear stress around the limb’s periphery as the socket is donned, restricting distal tip translation and contact.Despite the uniform pressure distribution targeted with the TSB design approach [4], asymmetries in strain field distributions were observed, in particular on the coronal plane, for example, with a tensile strain concentration near the medial anterior tibia (Figure 5 ‘dmt’). The same was observed for the PTB socket.

However, clear local distinctions between the strain field estimates were detectable for the two socket designs, most notably in sagittal plane axial and shear strain:

The TSB design socket produced some compressive axial strain in the distal ROI with median (10th–90th percentile ranges) of 0.58% (−3.67–4.58) on the sagittal plane. Conversely, the PTB socket design displayed lower axial strain magnitude than the TSB case in the distal ROI, especially visible in the sagittal plane, with median (10th–90th percentile ranges) of 0.12% (−2.50–3.45). This is consistent with the PTB design approach, which aims to achieve load proximal transfer, primarily at load-tolerant sites, and thereby off-load more sensitive and vulnerable sites, and bony prominences [5,6].The other notable difference was a change in the shear stain field directions and magnitudes on the sagittal plane, at the anterior distal tibial cut (Figure 5‘’dat’’), a bony prominence often linked with soft tissue sensitivity and vulnerability to damage. This shear strain gradient was produced despite a lack of limb–socket interface contact here for the PTB design (Figure 5 ‘’ig’’). The sagittal shear strain fields were both shifted to a larger magnitude in the anterior part of the ROI with median (10th–90th percentile ranges) of −2.08% (−8.88–14.69) for the PTB socket, compared to −5.79% (−11.41–0.64) for the TSB socket.

Correlation failed proximally around the knee and at the distal tip, due to the mentioned imaging artefacts and, respectively, sparser and denser sand content in these areas. Correlation was also lost in areas near the limb–socket interface, where there is no contact, including the distal tip in both sockets, and anterior to the distal tibial cut for the PTB socket.

## 4. Discussion

This study set out to develop an analogue transtibial amputated residual limb for volumetric strain estimation using DVC and assess its sensitivity to detect an example of the strain patterns typically observed in residual limbs during prosthetic socket use, representing double-leg standing. The analogue was used to illustrate a case study comparison between two common clinically-used socket design approaches. The estimated strain distributions from the DVC analysis identified key differences between the socket designs, providing an objective means to quantify these differences in both axial and shear strain distribution. 

The simplified bulk soft tissue analogue material was produced, including sand particles as fiducial markers for image correlation, and a compressive tangent modulus matched to literature reports for transtibial residual limb tissues. When incorporated into a full analogue residual limb mounted on an in situ loading rig, its DVC strain sensitivity was quantified as below 0.320% with a spatial resolution of approximately 3.5 mm. For strain magnitude, this gives the potential of high signal:noise ratio. The sand particles were well-distributed throughout the structure, except for some sparse regions within the knee joint space and a dense space at the distal tip, where ~5 mm particle sinking before full curing caused particle collection. From an imaging perspective, this was accepted as the sparse, proximal zones are above the functional socket area, so outside the main region of interest. However, this would elevate the modulus of the more dense distal tip, though again distal tip is not intended to bear load. Future work might investigate methods to further stabilise the sand distribution, such as rotating the mould during curing, or by accelerating curing chemically or thermally.

In the subsequent case study evaluation, the majority of the residual limb’s region of interest experienced estimated strain up to 10% for both axial and shear strains, which corresponded to the strain range for which the analogue material’s stiffness was designed. This is corroborated by finite element analyses, which give typical mean strain estimates of 3–15% for both principal tension, compression, and maximal shear, and strain peaks of 30–75% [15,17,38]. The estimated strain distribution lies below estimated thresholds for skeletal muscle damage from engineered tissue and rat models [39,40]. Strain predictions were consistent with the design intent for TSB [4] and PTB [5,6] socket types, and provided a noteworthy illustration of how sub-surface shear strain gradients—a marker of deep tissue injury risk—may be generated near bony prominences in the absence of interface pressure near the anterior distal tip of the tibia.

Whilst this new approach provides additional insights into these complex, non-linear load transfer mechanisms, limitations must be acknowledged. The presented model does represent a considerable simplification of the true anatomy and physiology of the residual limb. Accepting practical limitations, simplifications were made to match the most common approaches taken by the FEA community, such as the use of a single, nominally homogeneous soft tissue analogue structure [16]. The model permits limited knee flexion, where a fixed knee joint has been used in some FE approaches, and residual limb models or simulators [41,42], which were designed to allow assessment of the residuum–socket interface. Further, the material is more closely matched to transtibial residuum tissue properties than many modelling studies that continue to use a stiff, 200 kPa linear elastic model [16]. Diffusion tensor MRI may enable separate muscular structures to be modelled [43], although DVC would give erroneous strain measurements between separate structures as the continuum assumption would no longer hold. For similar reasons, this approach does not provide interface stress assessment, which has been a key focus of prior research, and to provide such data the analogue would need to be instrumented with interface pressure and shear stress sensors [44]. Interface displacement data would be enhanced by including markers in the socket, for DVC analysis; alternatively, reflective markers have been used to measure liner displacement inside a transparent check socket [45]. Finally, in this demonstration only a single residual limb is represented, so the case study does not provide a basis to draw general conclusions about socket designs, for which a statistically-representative range of limb shapes and sizes would be required. This approach sets aside other key factors like the stiffness and dynamics of associated componentry below the socket, such as the pylon and foot, and provides a constraint in that the fixed end of the bone and socket tip remain aligned on the loading axis, for test rig stability. This may not perfectly represent the freely-self-aligning in vivo scenario but is comparable between the two sockets. It is also acknowledged that the present study does not represent peak loading that would be expected during walking or other dynamic activity, but rather a quiet standing scenario. If strain can be assumed to correlate with load, this low load scenario is a harsher test for the methodology’s strain resolution and comparative power; however, a more complete picture of the potential range of loading conditions would be achieved with addition of moments to represent quasi-statically the key instants of the gait cycle [46].

This development provides an additional dimension to residual limb interface simulators [41,42], by providing a volumetric strain assessment. Non-invasive Digital Image Correlation (DIC) has also been demonstrated to offer some insights into residuum biomechanics, capturing in vivo the shape and free-deformation of the transtibial residual limb [47]. DVC may not yet be applicable in a clinical setting using CT imaging due to radiation dose and lack of contrasting pattern for correlation, and even standard field MRI is not collected routinely in these individuals. However, there may be value in using these techniques for in vitro comparative analysis of socket design parameters, and the validation of FE models.

## 5. Conclusions

An analogue residual limb was presented that provides a simplified model to estimate the soft tissue load distribution pattern generated by prosthetic sockets. These results indicate that physiological strain magnitudes can be observed, supporting the method’s use for comparative analysis of different socket designs, and an indication of design details or anatomic features associated with elevated deep-tissue injury risk [48].

## Figures and Tables

**Figure 1 materials-13-03955-f001:**
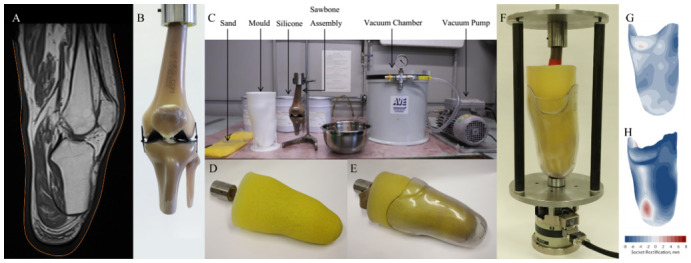
Producing analogue residual limb from MRI data (**A**). Residual bone models and simplified knee joint created, with nearest size-match to MR images (**B**). Residual bones suspended inside a mould; silicone and sand analogue material mixed, degassed, poured, and degassed again (**C**). Model donned with nylon stocking (**D**) and polyethylene terephthalate (PETE) check socket (**E**). Mounted on in situ loading rig (**F**). Indicative rectification maps from −8mm (blue) to +8mm (red) are given for the Total Surface Bearing (TSB) (**H**) and Patellar Tending Bearing (PTB) (**G**) sockets, calculated using the open source AmpScan shape analysis tool [37].

**Figure 2 materials-13-03955-f002:**
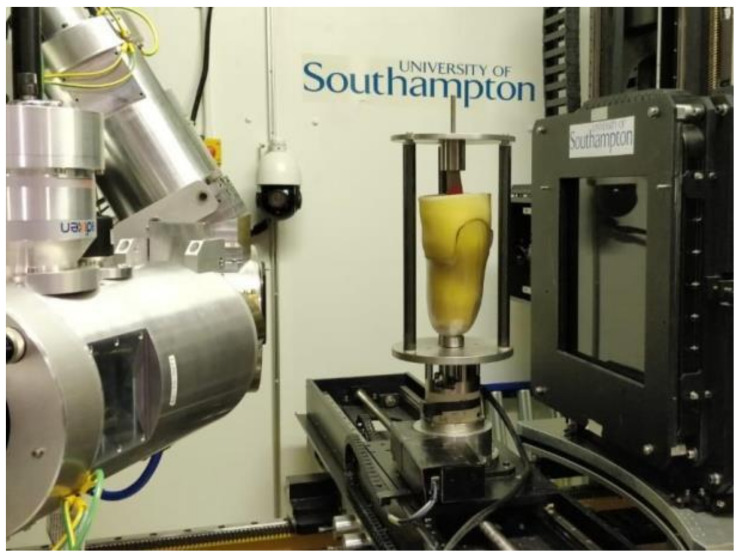
Analogue model and check socket mounted on the in situ loading rig inside the HUTCH microfocus CT scanner at µVIS X-ray Imaging Centre. Image shows X-ray source (left) and detector (right), with loading rig mounted on a rotate stage.

**Figure 3 materials-13-03955-f003:**
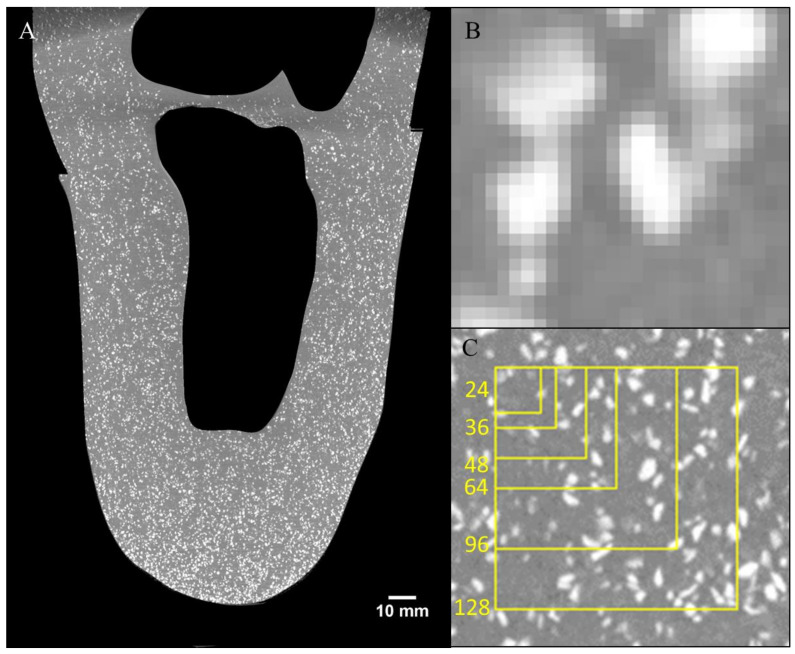
Example reconstructed CT section of a sagittal plane through the model, with bones and surrounding areas masked (**A**), magnified view of sand particle features compared to the 108 µm-cube voxels (**B**), and comparing subvolume size (yellow squares) from 24 to 128 voxels (2.6 to 13.8 mm) squares (**C**).

**Figure 4 materials-13-03955-f004:**
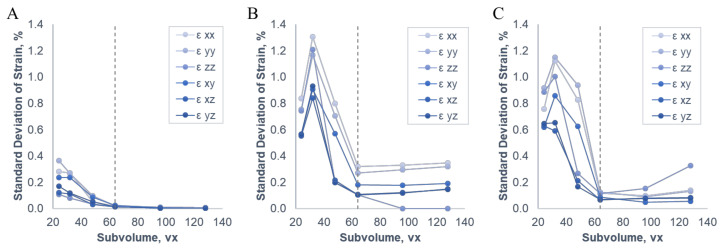
Noise study results for digital volume correlation (DVC) parameter selection. Plots of standard deviation of strain components vs. subvolume size, for the three unloaded noise study test conditions: repeat scans (**A**), nominal 1mm translation (**B**), and nominal 1% magnification (**C**). Selected 64 vx subvolume size indicated by dashed line.

**Figure 5 materials-13-03955-f005:**
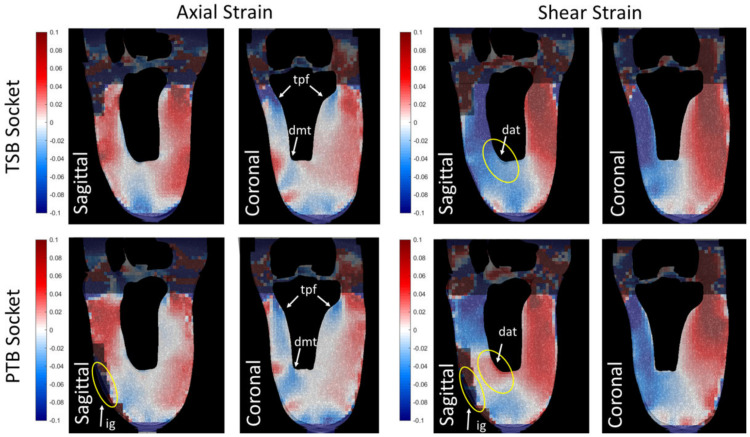
Example DVC axial- and in-plane shear strain maps for two socket designs: total surface bearing (TSB, top row) and patella tendon bearing (PTB, bottom row). Zones where correlation has failed are shaded. Annotation key: tpf: tibial plateau flare; dat: distal anterior tibia; dmt: distal medial tibia; ig: interface gap.

**Figure 6 materials-13-03955-f006:**
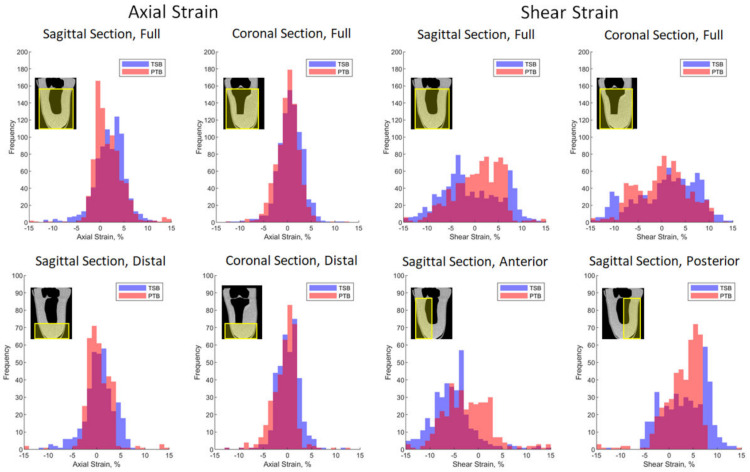
Example DVC axial- and in-plane shear strain distributions for TSB and PTB socket designs, in indicated regions of interest.

**Table 1 materials-13-03955-t001:** Scanning protocol.

Noise Scan ID	Description
1	Unloaded
2	Repeat unloaded
3	Repeat unloaded, translated axially ~1 mm
4	Repeat unloaded, translated towards X-ray source for magnification
5	TSB socket, nominally unloaded (<50 N)
6	TSB socket, single-leg stance axial load (400 N)
7	PTB socket, nominally unloaded (<50 N)
8	PTB socket, single-leg stance axial load (400 N)

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
