# Peer review of "Developing an Analogue Residual Limb for Comparative DVC Analysis of Transtibial Prosthetic Socket Designs"

_materials, 2020, doi:10.3390/ma13183955_

Round 1

Reviewer 1 Report

Using the non-destructive digital volume correlation, this study presents an experimental investigation to assess the strain field in a soft tissue material analog to a prosthetic device. The paper is well written with originality and  deserves publication after the following minor comments are addressed:

Page 2 line 67: To show how FE analysis can be used for the design or development of biomedical device. Please cite the following paper:

"Rinaudo A, Raffa GM, Scardulla F, Pilato M, Scardulla C, Pasta S. Biomechanical implications of excessive endograft protrusion into the aortic arch after thoracic endovascular repair. Comput Biol Med. 2015;66:235-241."  

Page 2 line 81: Indeed methods to derive deformation (engineering strain) from CT have been proposed. This method are based on the analysis of the displacement field of at least two cardiac phases collected from CT imaging. I kindly suggest to cite the following paper on this topic.

“Pasta S, Agnese V, Di Giuseppe M, et al. In Vivo Strain Analysis of Dilated Ascending Thoracic Aorta by ECG-Gated CT Angiographic Imaging. Ann Biomed Eng. 2017;45(12):2911-2920.”

Page 3 line 112: At which strain rate the load was applied?

Page 5 line 170: the material may show a viscoelastic effect. Thus, when the load is stopped for the image acquisition, the stress or strain can change over time. Could you please specify when the acquisition was done? immediately after the step or after a few time? did you observed change in stress/strain over time during a given step?

Page 5 line 176: In each subset, especially the smallest one, the speckle pattern is not so intense as it should be in DVC. There are many black areas with very few white spots, and this can result in a poor image quality for the correlation analysis. In this way, the subset size needs to be increased for the strain computation but the resulting strain distribution will be likely underestimated. Please provide more details on this aspect.

Author Response

Please see the attachment for full details, including response to Reviewer 2.

Reviewer 1:

Using the non-destructive digital volume correlation, this study presents an experimental investigation to assess the strain field in a soft tissue material analog to a prosthetic device. The paper is well written with originality and  deserves publication after the following minor comments are addressed:

Response: Very many thanks for the time you took to review our submission, and for these comments. We have responded and made changes in the Marked-Up version of the article, and in the following:

Page 2 line 67: To show how FE analysis can be used for the design or development of biomedical device. Please cite the following paper:

"Rinaudo A, Raffa GM, Scardulla F, Pilato M, Scardulla C, Pasta S. Biomechanical implications of excessive endograft protrusion into the aortic arch after thoracic endovascular repair. Comput Biol Med. 2015;66:235-241."  

Response: Thank you for this recommendation. We accept that the evidence here might be strengthened. As FEA for device development is highly established we would prefer, in this portion of the paper, to retain focus on literature from the most closely related area of limb prosthesis devices. Therefore, we reorganised the sentence to provide reference to the two seminal FEA socket design – interface stress studies, and save the mention of reference [10] til the end, justifying its inclusion due to computational efficiency sufficient for in-clinic use. (L67-70). 

Page 2 line 81: Indeed methods to derive deformation (engineering strain) from CT have been proposed. This method are based on the analysis of the displacement field of at least two cardiac phases collected from CT imaging. I kindly suggest to cite the following paper on this topic.

“Pasta S, Agnese V, Di Giuseppe M, et al. In Vivo Strain Analysis of Dilated Ascending Thoracic Aorta by ECG-Gated CT Angiographic Imaging. Ann Biomed Eng. 2017;45(12):2911-2920.”

Response: Thank you – this is a valuable additional alternative soft tissue strain assessment method, with in vivo capability, which we have added to the paper’s Introduction (L93-94). 

Page 3 line 112: At which strain rate the load was applied?

Response: we added the applied strain rate of 5%/sec. We conducted tests at 0.5%/sec and 50%/sec and saw minimal influence of strain rate on initial tangent modulus (0-10% strain range) (L116).

Page 5 line 170: the material may show a viscoelastic effect. Thus, when the load is stopped for the image acquisition, the stress or strain can change over time. Could you please specify when the acquisition was done? immediately after the step or after a few time? did you observed change in stress/strain over time during a given step?

Response: a good point, for which we have added an explanation. “Following preliminary tests, it was identified that a ~20 minute dwell period was required between loading and scanning, for viscoelastic stress relaxation effects to subside. A pre-determined displacement was applied to give the desired load after stress relaxation”. (L169-171). 

Page 5 line 176: In each subset, especially the smallest one, the speckle pattern is not so intense as it should be in DVC. There are many black areas with very few white spots, and this can result in a poor image quality for the correlation analysis. In this way, the subset size needs to be increased for the strain computation but the resulting strain distribution will be likely underestimated. Please provide more details on this aspect.

Response: Thanks for this query. Indeed, the residual limb model only includes sand particles in regions represented by the analogue soft tissue material. In Figure 3 (A) we have masked the image, as we masked the results – so we did not try to analyse strain in the black areas without speckles. The characterised speckle pattern subvolume size in Figure 3 (C) is taken from a representative region of the limb, and we add to the Figure 3 caption a key of the subvolume size range in mm terms, to support the currently-stated voxel terms, to aid understanding (L201). Further, the subsequent noise study data which support the study’s identified subvolume size used a large, representative volume of the model, capturing both the higher and lower-density speckle zones. We fully accept that above the tibia and socket brim-line in the sagittal view (Figure 3 (A)) the speckles are sparse and the unavoidable CT cone-beam artefact is visible, so for this reason we do not interrogate the strain above the tibia, and we acknowledge this in the Results (L190-193 and 243-6). In the response to Reviewer 2’s comments, we also made edits to the Discussion from L267-274.

Reviewer 2 Report

Review – Developing an analogue residual limb for comparative DVC analysis of transtibial prosthetic socket designs

This is a well written article, where the authors outline a novel and interesting approach to assessing changes to residual limb model geometry under loading. I only have minor comments or suggestions.

Introduction:

Line 52: Sentence starting ‘Several well-established theories…’ could be backed up with the references the authors have used to outline these approached later in the paper [31,32,33]. Suggest adding here also.

Line 59: The reference used in the sentence ending in ‘in their first year post amputation’ is giving re-presentation numbers specific to transfemoral amputation, which has a higher incidence of complications than the transtibial amputation that is used for a model in this study. This might be worth clarifying, or a reference specific to the transtibial population may be more relevant?

Materials and Methods:

Line 115: In the sentence starting ‘The compressive force’ there may be a typo ‘platen’ should read plate?

Authors mention the methods to fabricate the socket, but not whether any alignment procedure was followed? Was the bone held vertically and the rest of the model aligned on that basis or was some sort of replication of alignment of the socket by the prosthetist performed? Which regions were designed to be loaded and unloaded by the prosthetist in the PTB socket?

Results:

Line 178: The authors acknowledge particle sinking. It might be worth noting here whether or not this may have altered the tangent moduli since the sand was used to alter the mechanical behaviour of the silicone in the methods? Was anything done to test this effect?

Line 200: The authors use a region of interest (ROI) for analysis. I may have missed this, but why was only this region selected? This would mean that some of regions of loading for the PTB socket may have been missed more proximally or did the socket design ensure this wasn’t the case?

Line 207: The authors make a comment here about distal tip protection being achieved, but in line 230 they say distal tip correlation failed. Can the authors clarify here how they are able to comment on this region if the correlation failed? Is it that there was no contact between limb and socket during loading?

Line 229: Again the authors could comment here or in the methods about whether alignment between the two sockets was standardised to enable differences to be observed?

Discussion:

Line 224: The authors make a statement here about strain patterns observed during socket use, but only one static position assuming double stance was tested in this study. Could the authors perhaps phrase this as ‘sensitivity to detect an example of the strains that would occur in double leg standing’ or something similar?

Line 263: Would the authors consider adding some comments here or elsewhere in the discussion about the effect of the different sand distributions on behaviour? If it is important, could they make some suggestions for improving manufacture of future analogues eg: curing while the analogue is rotated about the longitudinal axis continuously or other?

Line 265: Are the authors justifying their choice of model as it links to the FEA research in the sentence starting ‘’It comprises…’? If so, consider stating this here more clearly as it seems a reasonable justification.

Line 268: ‘previously mentioned simulators’ – what are the authors referring to here, could you include a reference perhaps?

Line 270: Sentence ending ‘200kPa linear elastic model’ – do the authors have a reference for this model?

Line 275: Interface displacement data – Lenz et al have published a paper on a method using reflective markers to measure liner displacement under a check socket. This that could be worth including here to provide the reader some more guidance on a potential way to achieve it. It is called ‘Understanding Displacements of the Gel Liner for Below Knee Prosthetic Users’

Line 281: Adding moments or offsetting the orientation of the model could also give some interesting results. The authors could also mention in this section that the loading magnitude is not a ‘peak’ load that would be expected during walking, rather in quiet two-legged standing. Load could also be increased to assess effect on the models.

Author Response

Please see the attachment for more, including response to Reviewer 1.

Reviewer 2:

This is a well written article, where the authors outline a novel and interesting approach to assessing changes to residual limb model geometry under loading. I only have minor comments or suggestions.

 Response: Very many thanks to you too, for the time you took to review our submission, and for these comments. We have made corresponding changes in the Marked-Up version of the article, and respond as follows:

Introduction:

Line 52: Sentence starting ‘Several well-established theories…’ could be backed up with the references the authors have used to outline these approached later in the paper [31,32,33]. Suggest adding here also.

Response: a good idea – we have presented these references earlier as the reviewer suggests, now numbered [4-6]. (L53-54).

Line 59: The reference used in the sentence ending in ‘in their first year post amputation’ is giving re-presentation numbers specific to transfemoral amputation, which has a higher incidence of complications than the transtibial amputation that is used for a model in this study. This might be worth clarifying, or a reference specific to the transtibial population may be more relevant?

Response: this is a good point – the Haggstrom et al study does indeed refer to Transfemoral individuals only. However, the Pezzin et al study refers to a range of amputation levels, and finds the same average number of visits. We added a few words to mention that this number is not specific to transtibial cases (L60-61).

Materials and Methods:

Line 115: In the sentence starting ‘The compressive force’ there may be a typo ‘platen’ should read plate?

Response: Platen seems to be an accepted word as a load-applying plate, but for avoidance of confusion we edited this to ‘loading plate’ (L119).

Authors mention the methods to fabricate the socket, but not whether any alignment procedure was followed? Was the bone held vertically and the rest of the model aligned on that basis or was some sort of replication of alignment of the socket by the prosthetist performed? Which regions were designed to be loaded and unloaded by the prosthetist in the PTB socket?

Response: The bone was held vertically with respect to the limb external shape axis identified from the MRI scan, for which the reader might see a flat-bottomed support to the white mould in Figure 1 (C); therefore the limb model was built with reference to the patient-specific medical imaging. The socket was allowed to take up an equilibrium position relative to the limb in donning, and since the sockets are grossly similar (only their local interference fit pattern differs) there were no differences in applied limb-socket constraint. The model was aligned in the test rig so that the centre of the cut end of the femur aligned vertically with the distal-most tip of the socket, and a cylindrical support was used to transfer load from the displacement actuator to the socket, thus allowing the socket to rotate with near-freedom. Therefore, the only constraints applied were the loading axis direction, and setting the socket tip to remain near to this axis under load. Manuscript is edited to reflect this (L147-151)

With regard to your question about the PTB socket design, we have added two further panels to Figure 1, (G) and (H) which provide the socket rectification maps, and where a blue colour indicates design for loading, and red indicates design for unloading (L156-157).

Results:

Line 178: The authors acknowledge particle sinking. It might be worth noting here whether or not this may have altered the tangent moduli since the sand was used to alter the mechanical behaviour of the silicone in the methods? Was anything done to test this effect?

Response: This is true. From an imaging perspective, this was accepted as the sparse, proximal zones are above the functional socket area, so outside the region of interest, and the denser region would only be easier to correlate. However, this would elevate the modulus of the more dense distal tip. This was accepted again because the distal tip is not intended to bear load, although the tangent modulus of more and less dense portions of the limb were not tested. We have added a comment to the Discussion to explain this. (L271-272).

Line 200: The authors use a region of interest (ROI) for analysis. I may have missed this, but why was only this region selected? This would mean that some of regions of loading for the PTB socket may have been missed more proximally or did the socket design ensure this wasn’t the case?

Response: the ROI employed in the results extraction was large, encompassing almost all the analogue soft tissue material within the socket. The masked regions of Figure 5 have been updated for consistency with Figure 6, to reflect this. Although not responsible for the majority of socket load transfer in stance (more intended for suspension) there are some more proximal medial and lateral portions of soft tissue which are outside this ROI, and where reliable strain assessment was not possible, and this was suspected to be due to the positioning of the model inside the CT cone-beam artefact. If this region were of interest, the tests could be repeated, moving this region away from these unavoidable artefacts.

Line 207: The authors make a comment here about distal tip protection being achieved, but in line 230 they say distal tip correlation failed. Can the authors clarify here how they are able to comment on this region if the correlation failed? Is it that there was no contact between limb and socket during loading?

Response: this is correct, our writing was imprecise here and we have made a small edit to the text. We are focusing here on the distal soft tissue between the tip of the tibia and the tip of the limb. Only a small (~5mm) slice of the distal tip soft tissue is lost to correlation failure, but the differences in soft tissue loading extend right up to the residual bone fragment, and it is this location we are discussing. We made an edit to note it is the ‘distal soft tissue’ (L220) in general which is protected, not just the tip, which admittedly we cannot see – but it is that tissue near to the bony prominence which is of greatest concern with respect to deep tissue injury.

Line 229: Again the authors could comment here or in the methods about whether alignment between the two sockets was standardised to enable differences to be observed?

Response: as we attempted to provide the same, minimal constraint to the sockets, whilst ensuring stability in the test rig, these results should be a comparable representation of the two sockets’ load transfer mechanisms. Please see our edit in the Discussion (L306-309).

Discussion:

Line 244: The authors make a statement here about strain patterns observed during socket use, but only one static position assuming double stance was tested in this study. Could the authors perhaps phrase this as ‘sensitivity to detect an example of the strains that would occur in double leg standing’ or something similar?

Response: this is a good suggested edit, which we have made, almost word-for-word (L256-257).

Line 263: Would the authors consider adding some comments here or elsewhere in the discussion about the effect of the different sand distributions on behaviour? If it is important, could they make some suggestions for improving manufacture of future analogues eg: curing while the analogue is rotated about the longitudinal axis continuously or other?

Response: Again a nice suggestion, which we have included. (L272-274).

Line 265: Are the authors justifying their choice of model as it links to the FEA research in the sentence starting ‘’It comprises…’? If so, consider stating this here more clearly as it seems a reasonable justification.

Response: we flipped the sentence in response to this comment, and hopefully the logic is now more clear, as suggested. (L287-290).

Line 268: ‘previously mentioned simulators’ – what are the authors referring to here, could you include a reference perhaps?

Response: Apologies, this was a continuity error after the mention of ‘simulators’ (refs [43] and [44] were moved to later in the Discussion section. We have reorganised the chapter with edits at (L291-293 and L315-317).

Line 270: Sentence ending ‘200kPa linear elastic model’ – do the authors have a reference for this model?

Response: a reference added to a Review paper which traces the uses of several simulation studies back to the model’s original source. We could reference that one original model’s source, if the Reviewer prefers, but we judged the Review paper would tell the whole story. (L295).

Line 275: Interface displacement data – Lenz et al have published a paper on a method using reflective markers to measure liner displacement under a check socket. This that could be worth including here to provide the reader some more guidance on a potential way to achieve it. It is called ‘Understanding Displacements of the Gel Liner for Below Knee Prosthetic Users’

Response: thanks for mentioning this alternative and clinically-useable method. We have added the reference as suggested (L301-302).

Line 281: Adding moments or offsetting the orientation of the model could also give some interesting results. The authors could also mention in this section that the loading magnitude is not a ‘peak’ load that would be expected during walking, rather in quiet two-legged standing. Load could also be increased to assess effect on the models.

Response: another good observation. We argued this presents a harsher test to the method’s strain resolution, as these quiet conditions will give a lower strain signal, but fully acknowledge that a wider range of higher loads should be considered when applying the methodology. We made an edit as suggested, highlighting that this is not a peak loading scenario, and our motivation (beyond the practical simplicity of starting with the simpler, axial load case). (L309-312).

Thank you again for these thorough and helpful comments, which have strengthened the paper’s detail and communication!
